# Observation of full-parameter Jones matrix in bilayer metasurface

Yanjun Bao [1]✉, Fan Nan[1], Jiahao Yan[1], Xianguang Yang [1], Cheng-Wei Qiu [2]✉ & Baojun Li[1]✉

Metasurfaces, artificial 2D structures, have been widely used for the design of various functionalities in optics. Jones matrix, a 2×2 matrix with eight parameters, provides the most complete characterization of the metasurface structures in linear optics, and the number of free parameters (i.e., degrees of freedom, DOFs) in the Jones matrix determines the limit to what functionalities we can realize. Great efforts have been made to continuously expand the number of DOFs, and a maximal number of six has been achieved recently. However, the realization of the ultimate goal with eight DOFs (full free parameters) has been proven as a great challenge so far. Here, we show that by cascading two layer metasurfaces and utilizing the gradient descent optimization algorithm, a spatially varying Jones matrix with eight DOFs is constructed and verified numerically and experimentally in optical frequencies. Such ultimate control unlocks opportunities to design optical functionalities that are unattainable with previously known methodologies and may find wide potential applications in optical fields.

The design of optical structures with arbitrary functionalities has always been an ultimate dream for people in optics. However, the number of the degrees of freedom (DOFs) of light control by the implemented optical structure itself determines the limit to what functionalities that can be realized. In linear optics, the optical structures can be completely characterized by a spatially varying 2 × 2 Jones matrix[1], which contains of eight parameters, therefore indicating a maximal number of eight DOFs (otherwise known as the free parameters that can be arbitrarily varied) for all linear structures in nature. Obviously, the more free parameters in the Jones matrix can be varied, the diverse functionalities we can achieve. The highest eight DOFs (full free parameters) represent the most general control in optics and are the foundation to realize the most complex optical functionalities. So, the question arises that how to expand the number of DOFs in Jones matrix, even to the ultimate eight DOFs?

Metasurfaces, which consist of a monolayer of planar structures, provide a suitable platform for spatially varying light control in sub-wavelength scales[2–4]. The metasurfaces have been extensively explored in the past ten years for diverse functionalities, such as anomalous refraction[4–8], hologram[9–16], metalens[17–22], vortex beam[23–26], etc. In fact, almost all these functionalities can be categorized into the different DOFs in the Jones matrix, which manifests the process of continuous endeavors to expand the number of DOFs. For example, the $x$-polarized anomalous refraction with $y$-polarized incidence[4] can be attributed to the phase control of the $J_{12}$ component of the Jones matrix, i.e., one DOF. The polarization-control dual holographic images[16] is enabled by the independent phase control of the two diagonal entries of the Jones matrix, i.e., two DOFs. An application of four DOFs (the amplitude and phase terms of both $J_{12}$ and $J_{22}$ components in the Jones matrix) is to generate arbitrary amplitude, phase, and polarization distributions[23]. Due to the mirror symmetry, the Jones matrix of planar structure is symmetric and thus has an upper limit number of six DOFs[27], which is constructed with metasurface recently[28,29]. To break the mirror symmetry, multi-layer design is necessary. Recently, Yuan et al.[30] proposed a five-layer metallic structure to independently control the phases of the four components of the Jones matrix (circular polarization base) in microwave reign (i.e., four DOFs). Although great achievements have been made in expanding the number of DOFs, the goal with ultimate

[1]Institute of Nanophotonics, Jinan University, Guangzhou 511443, China. [2]Department of Electrical and Computer Engineering, National University of Singapore, Singapore 117583, Singapore. ✉e-mail: yanjunbao@jnu.edu.cn; chengwei.qiu@nus.edu.sg; baojunli@jnu.edu.cn

eight DOFs in Jones matrix has not yet been realized. The construction of such Jones matrix, especially operated in optical frequencies, is of great importance and meaningful to the field of optical design.

Here, we proposed and experimentally demonstrated an arbitrary spatially varying Jones matrix with eight DOFs in optical frequencies by using a bilayer metasurface to break the mirror symmetry of planar structures (Fig. 1a). The Jones matrix distributions of the two single layers in the bilayer structure are calculated based on gradient descent optimization algorithm, and the optimized results can agree well with any designed target distributions. The Jones matrix with eight DOFs provides unparalleled control of light. One example is that we can impose arbitrary and independent amplitude and phase control on any set of two polarizations (Fig. 1b). Most importantly, there are no restrictions on the input and output polarizations. In comparison, previously reported Jones matrix with three DOFs can impart independent phase[31] (or amplitude[32]) control on orthogonal polarizations only, and the output polarizations must be the same as the input ones with flipped handedness (or mutually conjugate), i.e., the output two polarizations are orthogonal with each other, too. Following work with six DOFs[33] extends the above light control and can impart both independent amplitude and phase control, but exhibits the same restrictions on the input and output polarizations as that with three DOFs. The comparison in Fig. 1c highlights the unique and versatile control with eight DOFs. In addition, the bilayer design introduces another rotation DOF, which is utilized to realize polarization-rotation multiplexed holography reaching up to 16 independent functionalities.

## Results

Figure 1a illustrates the schematic view of our designed structure with eight DOFs in the Jones matrix, which consists of two layers of

metasurfaces with a separation $z$ between them. Such bilayer metasurface design approaches have been previously proposed for various optical functionalities[34–44]. Here, the Jones matrix of each of the two layers $J^1(x_1,y_1)$ and $J^2(x_2,y_2)$ is symmetric due to mirror symmetry and assumed to have upper limit of six DOFs to provide enough design freedom. When light impinges from the bottom of the bilayer metasurface, it firstly passes through the first layer, then propagates over a distance $z$ in gap (homogenous environment) and finally passes through the second layer. The incident and output Jones vectors through the bilayer metasurface can be characterized by a spatially varying $2 \times 2$ Jones matrix $J$, which represents the optical properties of the whole optical system. In the following, we refer to it as equivalent Jones matrix (EQJM) in order to distinguish from that of the two single layers. The mn$th$ ($m, n = 1, 2$) component of the EQJM can be written as (see Supplementary section 1)

$$J_{mn}(x_2,y_2) = \sum_{q=1,2} J^2_{mq}(x_2,y_2) \iint_{x_1,y_1} J^1_{qn}(x_1,y_1) \cdot f(x_2 - x_1, y_2 - y_1, z) dx_1 dy_1$$

(1)

where $J^1_{qn}$ and $J^2_{mq}$ are the $qn$th and $mq$th component of the Jones matrix of the first and second layer, respectively, $f(x_2 - x_1, y_2 - y_1, z) = \frac{1}{2\pi} \frac{\exp(ikr)}{r} \frac{z}{r} (\frac{1}{r} - i\frac{2\pi}{\lambda})$ is the Rayleigh–Sommerfeld impulse response, $r = \sqrt{(x_1-x_2)^2 + (y_1-y_2)^2 + z^2}$, $i$ is the imaginary unit, $\lambda$ is the wavelength and $z$ is the distance between the two layers. Note that the Jones matrix of a single layer is symmetric and therefore $J^1_{12}(x_1,y_1) = J^1_{21}(x_1,y_1)$ and $J^2_{12}(x_2,y_2) = J^2_{21}(x_2,y_2)$.

We aim to find the Jones matrix values of the two single layers ($J^1_{ij}(x_1,y_1)J^2_{ij}(x_2,y_2)$, $ij = 11, 12$ (21), 22) to design an arbitrary spatially varying EQJM with eight DOFs. The analytical expression in Eq. (1)

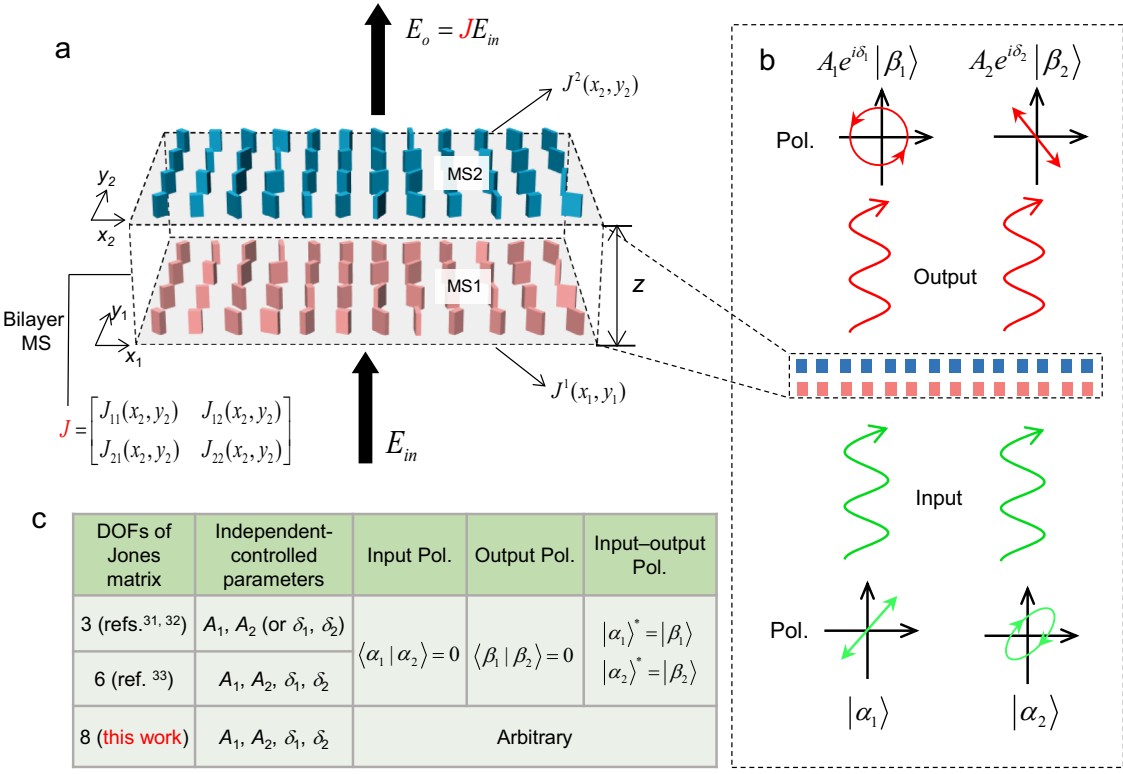

**Fig. 1 | Jones matrix with eight DOFs and an application example of advanced light control. a** Schematic view of a bilayer metasurface with eight DOFs in the Jones matrix. The incident and output Jones vectors $E_{in}$ and $E_o$ are connected by 2×2 equivalent Jones matrix $J$. The two layer metasurfaces are separated by a distance $z$. The Jones matrix of each single layer is symmetric and endowed with six DOFs. MS: metasurface. **b** Schematic of the bilayer metasurface for independent amplitude

and phase control of arbitrary set of two polarizations. The incident two arbitrary polarizations $|\alpha_1\rangle$ and $|\alpha_2\rangle$ can be transformed into arbitrary output polarizations $|\beta_1\rangle$ and $|\beta_2\rangle$ with independent complex-amplitude $A_1 e^{i\delta_1}$ and $A_2 e^{i\delta_2}$, respectively. There are no constraints imposed on the input and output polarizations. **c** Comparison of the independent controlled parameters and polarization constraints between metasurfaces with different DOFs in Jones matrix.

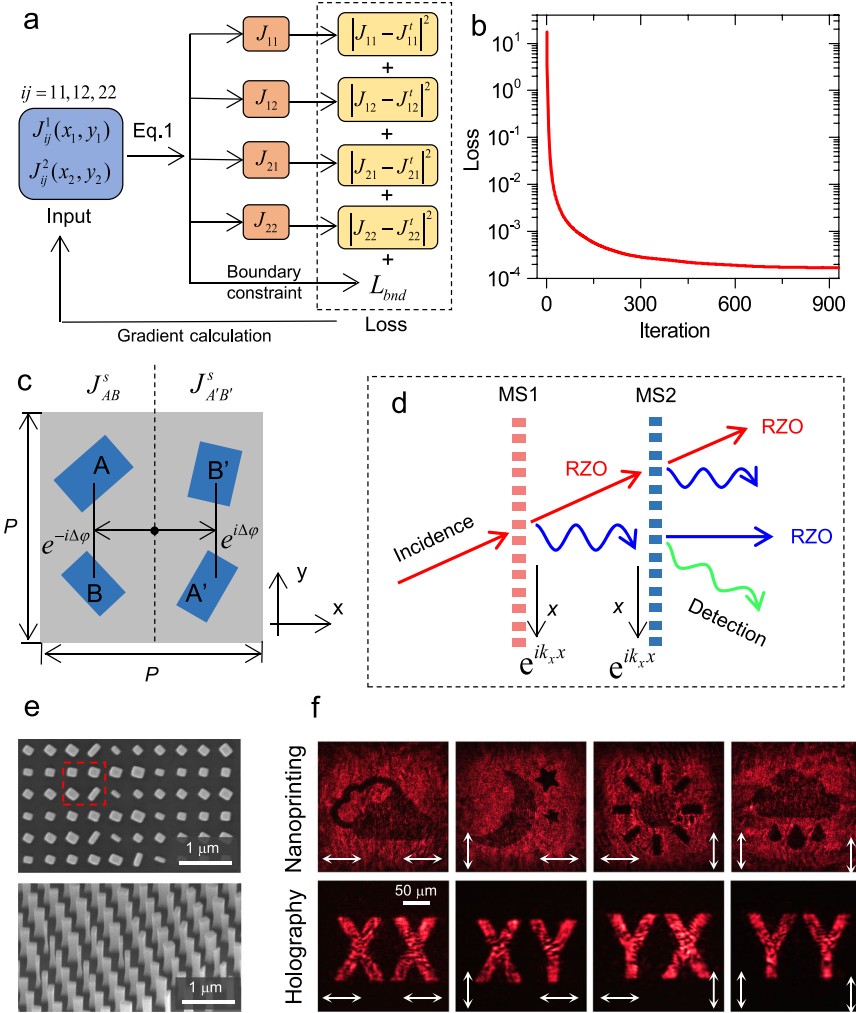

**Fig. 2 | Design of Jones matrix with eight DOFs and measurement results. a** Flow chart of the gradient descent optimization algorithm for design of Jones matrix with eight DOFs. The input variables are the Jones matrix components of the two single layer metasurface ($J^1_{ij}(x_1, y_1)$ $J^2_{ij}(x_2, y_2)$, $ij = 11$, 12 (21), 22). The defined loss includes two parts: the mean of the absolute squared differences of EQJM components between the prediction ($J_{ij}$) and target ($J^t_{ij}$), and a boundary constraint loss $L_{bnd}$. **b** Loss value as a function of the iteration number in the gradient descent optimization algorithm. **c** Unit cell of the single layer metasurface to construct Jones matrix with six DOFs. Each unit contains of four dielectric nanopillars, and the two sets of nanopillars $AB$ and $A'B'$ can individually and independently construct Jones matrix with six DOFs. Under oblique incidence or oblique scattering, detour phases of $e^{-i\Delta\varphi}$ and $e^{i\Delta\varphi}$ are imposed on the two sets of nanopillars $AB$ and $A'B'$. To compensate the detour phases, the Jones matrixes of the two set of nanopillars are chosen as $J^s_{AB} = e^{i\Delta\varphi}J$ and $J^s_{A'B'} = e^{-i\Delta\varphi}J$ to constructively generate the designed Jones matrix $J$ for the unit cell. The period of the squared unit pixel is $P = 800$ nm. **d** Schematic of light propagation through the bilayer metasurface with the oblique incidence-oblique detection measurement strategy. For each layer, the scattering light (indicated by wavy line arrows) is deflected by applying a gradient phase distribution $e^{ik_x x}$ on it and the residual zero order maintains its propagation direction as the incident one. With such measurement, the influence of the residual zero orders can be totally eliminated in the detection. RZO: residual zero order. **e** Scanning electron microscopy (SEM) images of the fabricated metasurface (partial view). The dashed red square indicates one unit pixel with individually designing the two sets of nanopillars. **f** Experimental results of the measured nanoprinting and holographic images. The incident and analyzed polarizations are indicated at lower left and lower right of each panel. All panels share the same scale bar in the lower left one.

allows the convenient use of gradient descent optimization algorithm to obtain the optimized solutions (Fig. 2a and see Supplementary section 2 for details). Besides the conventional absolute-mean-squared-error loss, we also add a boundary constraint loss $L_{bnd}$ to ensure that this algorithm converges to solutions that are inside the valid domains. The specific form of the boundary loss is determined based on the metasurface unit design, which will be introduced in the following. A detailed discussion of this boundary loss is provided in supplementary section 3.

To verify our approach, a target EQJM is designed with its amplitude and phase distributions chosen to present four nanoprintings (trinary intensity images of weather symbol) and four holographic images (letter strings "XX", "XY", "YX" and "YY") encoded in its four components (The details can be found in supplementary section 4). Figure 2b plots the loss value obtained from the algorithm as a function of the iteration number, which decays rapidly and reaches convergence after nearly 600 iterations. The optimized results of the EQJM, including both the amplitude and phase distributions, show good agreements with the targets (supplementary fig. 2). Only a slight deviation of the magnitude occurs near the boundaries, where the sharp truncation of the magnitude contains high angular spectrum frequencies that extend beyond our design.

The two single-layer metasurfaces with six DOFs of Jones matrix can be constructed by multi-element unit design[28,29]. Such multi-element unit design (diatomic, tetratomic, et al.) has shown unique advantages in increasing the DOFs of controlling light[45–47]. In this case, the Jones matrix

is decomposed as the summation of the ones of the individual elements. We consider the element with rectangle dielectric nanopillars of silicon on glass, which have a higher refractive index than the surrounding environment air. The Jones matrix of the single layer metasurface with two-element (nanopillars $A$ and $B$, Fig. 2c) unit is given by

$$J_{AB}^s = \begin{bmatrix} J_{11}^s J_{12}^s \\ \sim J_{22}^s \end{bmatrix} = R(-\theta_A)\begin{bmatrix} e^{i\varphi_1} & 0 \\ 0 & e^{i\varphi_2} \end{bmatrix}R(\theta_A) + R(-\theta_B)\begin{bmatrix} e^{i\varphi_3} & 0 \\ 0 & e^{i\varphi_4} \end{bmatrix}R(\theta_B) \quad (2)$$

where $\varphi_1$ ($\varphi_3$) and $\varphi_2$($\varphi_4$) are the phase shifts imposed on the light linearly polarized along the fast and slow axes of nanopillar $A$ ($B$), $\theta_{A,B}$ are the rotational angles of the two nanopillars and $R(\theta_{A,B})$ is the 2 × 2 rotation matrix. Clearly, not all symmetric Jones matrix can be decomposed as that in Eq. (2). A sufficient precondition of Eq. (2) is $|J_{11}^s| + |J_{12}^s| \leq 2$ and $|J_{12}^s| + |J_{22}^s| \leq 2$. The details of the derivations are provided in supplementary section 3. Therefore, a boundary constraint loss is added in the gradient descent algorithm to ensure that the input Jones matrix values always fall inside the above domains.

The phase shift values $\varphi_1$ ($\varphi_3$) and $\varphi_2$($\varphi_4$) are associated with the transverse dimensions (length and width in $xy$ plane) of the nanopillar. Full wave finite-difference time-domain (FDTD) simulations are performed, and a library of the transmission magnitudes and phase shifts dependent on the transverse dimensions of the nanopillar with incident $x$- and $y$- polarizations is built. With such databases, any $\varphi_1$ ($\varphi_3$) and $\varphi_2$($\varphi_4$) combinations ranging from 0 to $2\pi$ can be achieved by properly selecting the transverse dimensions of the nanopillars (see details in Supplementary section 4). As the optical response of nanopillar remains almost unchanged with the incident angle, such a library can be used for cases with oblique incidences (Supplementary Fig. 5).

The two nanopillars $A$ and $B$ are arranged vertically along $y$ axis, and to create a square pixel, a simple way is to duplicate the set of nanopillars $AB$ to $A'B'$ and distribute them uniformly within the pixel (Fig. 2c). This treatment is appropriate under normal incidence and normal scattering. For oblique incidence or oblique scattering (along $x$ direction), the introduced detour phases of the two sets of nanopillars can cause a major reduction of efficiency and deteriorate the optical performance (see discussion in Supplementary section 7).

When the incident light passes through the first layer metasurface, besides the transmitted scattering field, the unwanted residual zero-order light also imposes on the second layer. The two beams then pass through the second layer metasurface and respectively generate both the scattered field and residual zero order. Here the field of interest only is the scattering from the second layer illuminated by the scattering from the first layer. The residual zero-order light may strongly affect the measurement when its magnitude is comparable to that of the designed images.

We apply an oblique incidence-oblique detection measurement strategy, as shown in Fig. 2d. Here, the light is obliquely incident on the first layer, and the transmitted scattering is bended towards the normal direction by applying a gradient phase distribution $e^{ik_x x}$ ($k_x = 0.3k$, $k$ is the wavevector of the light in air) on the first layer. The same gradient phase distribution is also added on the second layer, which diffracts the incident normal scattering to an oblique direction. The influence of the residual zero orders can be totally eliminated if only the obliquely diffracted scattering is collected for imaging. In each pixel design, the Jones matrix of the nanopillars $AB$ and $A'B'$ are set as $e^{i\Delta\varphi}J$ and $e^{-i\Delta\varphi}J$ to compensate the detour phases, where $\Delta\varphi$ is the detour phase arisen from the oblique incidence or oblique scattering and $J$ is the designed Jones matrix value for the pixel. The advantages of the above measurement strategy are demonstrated (see Supplementary Fig. 6) by performing the full-wave FDTD simulations of the realistic bilayer metasurface under different optical measurement setups (normal incidence-normal detection, oblique incidence-normal detection, and oblique incidence-oblique detection). The details of the simulation processes are provided in supplementary section 6. It is

noted that the image quality of the holographic image maintains almost the same fidelity under all measurement setups. This is mainly because the holographic image is designed to be highly focused in far-field with its magnitude much larger than the diffracted background noise of the residual zero orders. In addition, a comparison between the consideration of the detour phase in the unit pixel design and without is shown in Supplementary Fig. 10. We also investigate the optical performance on the alignment of the two layers, which reveals that the images can be recognized with translational movement shift less than about 5 µm (Supplementary Fig. 11).

To set up the bilayer metasurface, we utilized two metasurfaces on separated substrates and cascaded them front-to-front to maintain a homogeneous environment (air) between them. The two metasurface samples are fabricated on 600 nm height crystal silicon layer that is transferred on glass substrate. The patterns are then defined by electron beam lithography (EBL) and reactive ion etching (RIE) process. The details of the fabrication procedure are outlined in the Methods. Figure 2e shows the SEM images of the metasurfaces, where the different designs of nanopillar sets of $AB$ and $A'B'$ within the pixel are observed. The oblique view of the metasurface shows the smooth sidewall profiles of these nanopillars. To measure the optical images along the oblique direction, we carry out a spatial filtering process in the Fourier plane. The Fourier plane of the objective lens (usually lies inside its barrel) is taken outside by adopting two lenses, which is then readily available for the filtering process. The details of the optical measurement are shown in Methods and Supplementary section 9.

The experimental results of the measured optical images with different combinations of the incident and analyzed polarizations are shown in Fig. 2f, which have good agreements with the designed targets. Although the measurement of the nanoprintings can be easily influenced by optical setups, the trinary intensity distributions are clearly seen for all four nanoprintings, demonstrating the accurate amplitude control of the EQJM. Due to the phase variations (arising from oblique incidence-oblique detection), the measured nanoprinting images exhibit fringe patterns, which are also verified in the simulations (Supplementary Fig. 6).

The efficiencies of the holographic images are measured to be 14.2% and 13.5% for $x$-polarized and $y$-polarized incidences, respectively (Supplementary section 9). Although the operation wavelength (808 nm) has been chosen to minimize the absorption loss of the silicon, the measured efficiency is somewhat lower than the common dielectric metasurfaces. The reason lies in two aspects. One is that the Jones matrix is designed with spatially varied amplitude distributions (the magnitudes in certain areas are nearly zero), which reduces the transmission through the metasurface. As a result, the efficiency of each single layer in our design is lower than that of the pure phase-only hologram. The other is that the efficiency of the bilayer metasurface is the product of the two single-layer ones, which reduces the efficiency value further. Therefore, the measured efficiencies after considering the two factors still meet the expected values of dielectric metasurfaces.

Next, we aim to demonstrate the proposed light control enabled by the eight DOFs that is mentioned in Fig. 1b. It is worth to mention that a recent theoretical work[48] has shown that the binary metasurface can implement arbitrary unitary polarization transformation, but lacks the control of the amplitude DOFs and is limited to orthogonal polarizations only. Let the two incident arbitrary polarizations be given by Jones vectors $|\alpha_1\rangle = \begin{bmatrix} \gamma_1 \\ \gamma_2 \end{bmatrix}$ and $|\alpha_2\rangle = \begin{bmatrix} \gamma_3 \\ \gamma_4 \end{bmatrix}$. The metasurface transforms the two input polarizations to output polarizations $|\beta_1\rangle = \begin{bmatrix} \chi_1 \\ \chi_2 \end{bmatrix}$ and $|\beta_2\rangle = \begin{bmatrix} \chi_3 \\ \chi_4 \end{bmatrix}$ with independent complex-amplitude control of $A_1 e^{i\delta_1}$ and $A_2 e^{i\delta_2}$, respectively (Fig. 1b). Note that we do not impose any constraints on the input and output polarizations. The Jones matrix of the

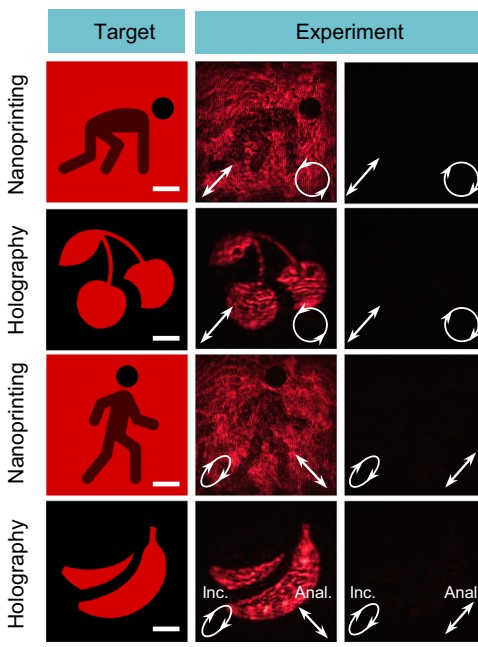

**Fig. 3 | Experiment demonstration of the independent amplitude and phase control of arbitrary set of two polarizations based on Jones matrix with eight DOFs.** The figure shows the target and measured nanoprintings and holographic images. The incident and analyzed polarizations are indicated in lower left and lower right corners of each panel. Scale bar: 50 μm.

metasurface should simultaneously satisfy

$$J|\alpha_1\rangle = A_1 e^{i\delta_1}|\beta_1\rangle \tag{3}$$

and

$$J|\alpha_2\rangle = A_2 e^{i\delta_2}|\beta_2\rangle \tag{4}$$

The Jones matrix can be directly extracted as

$$J = \begin{bmatrix} A_1 e^{i\delta_1}\chi_1 & A_2 e^{i\delta_2}\chi_3 \\ A_1 e^{i\delta_1}\chi_2 & A_2 e^{i\delta_2}\chi_4 \end{bmatrix} \begin{bmatrix} \gamma_1 & \gamma_3 \\ \gamma_2 & \gamma_4 \end{bmatrix}^{-1} \tag{5}$$

Obviously, we can always use the bilayer metasurface to construct a Jones matrix that satisfies Eq. (5) only if the input two polarizations are not exactly the same. It is worth mentioning that if only one of the two conditions (Eqs. 3 and 4) is satisfied, i.e., converting arbitrary polarization into another polarization state with independent amplitude and phase control, the Jones matrix requires a minimal number of four DOFs[49]. For demonstration, we choose nonorthogonal linear polarization ($\gamma_1 = \frac{\sqrt{2}}{2}, \gamma_2 = \frac{\sqrt{2}}{2}$) and elliptical polarization ($\gamma_3 = \frac{\sqrt{2}}{2}, \gamma_4 = \frac{\sqrt{2}}{2}e^{i\pi/3}$) as the input ones, which are transformed into another nonorthogonal circular polarization ($\chi_1 = \frac{\sqrt{2}}{2}, \chi_2 = -\frac{\sqrt{2}}{2}i$) and linear polarization ($\chi_3 = \frac{\sqrt{2}}{2}, \chi_4 = -\frac{\sqrt{2}}{2}$), respectively. The amplitude and phase control are demonstrated by designing two nanoprintings and two holographic images (first column in Fig. 3). When the incident and analyzed polarizations are set as the designed ones, we can observe clear nanoprintings and holographic images in the measurement (second column in Fig. 3), agreeing well with the targets. The output polarizations are verified from the observed almost dark images when switching the analyzed polarizations to the orthogonal ones (third column in Fig. 3).

The bilayer metasurface design introduces another DOF, the rotation angle between the two layers (Fig. 4a). It can be imagined that when the two layer metasurfaces rotate with respect to each other, the whole structure will exhibit different optical responses, which are that we want to control. The proposed bilayer metasurface is a good platform for multifunctional control as it provides enough DOFs ($12N^2$, $N$ is the pixel number along one direction) for design. We consider four cases with rotational angles $\phi = 0°$, $90°$, $180°$, and $270°$, and in each case, four independent functionalities are designed for each of the four different combinations of the incident and analyzed polarizations, i.e., a total of 16 polarization-rotation multiplexed functionalities. Here, we only consider the holographic functionality, as it is robust to the measurement conditions (see Supplementary Fig. 6). The loss is defined as the summation of the 16 squared differences between the magnitudes the predicted holographic images and the targets, plus the boundary loss (see details in Supplementary section 10). Then the gradient descent optimization algorithm is used to retrieve the Jones matrix values of the two single layers. To avoid the overlapping of the holographic image and the possible residual zero order, we applied the oblique incidence-normal detection measurement strategy. Figure 4b display our measured 16 holographic images under different combinations of the four rotation angles, two incident polarizations and two analyzed polarizations. All the measured images have high fidelity and have good agreement with the calculated optimized results (Supplementary Fig. 14). More importantly, the measured results almost have no cross-talk between any two holographic images, demonstrating the full independent multifunctionalities.

Although the whole device in the demonstration is not that compact (the gap distance between the two layers is set as 150 μm for the measurement convenience), our design strategy is general, and can be extended to gap distance towards several wavelengths (e.g., 5 μm), as shown in Supplementary section 11. For certain applications, it may be more convenient to integrate the two layer metasurfaces into monolithic system[38–41], but at the cost of disabling the rotation DOF. In addition, the operation bandwidth of current implementations is over 60 nm (Supplementary Fig. 16). It is noteworthy that although the demonstration was designed at 808 nm, which is aimed to avoid the large optical loss of silicon, our approach applies to short wavelengths in the visible range and can maintain performance with other lossless materials, such as TiO$_2$.

## Discussion

In conclusion, we have cascaded two single layer metasurfaces with six DOFs in the Jones matrix to construct a spatially varying Jones matrix with full parameters of eight DOFs, the maximal number allowed in nature. This represents a significant advance in the state-of-the-art for light control. Enabled by the eight DOFs of light control, we have demonstrated functionalities with independent amplitude and phase control of arbitrary set of two polarizations, without any constraints on the incident and output polarizations. In addition, we have investigated the DOF of rotation in the bilayer metasurface and demonstrated polarization-rotation multiplexed holography reaching up to 16 independent functionalities. We believe that our proposed design strategy of eight DOFs in Jones matrix offers a generalize method towards arbitrary control of light and may find applications that are not attainable with conventional methods.

## Methods

### Sample fabrication

A commercial SOI wafer with a 1200 nm-thickness of device layer is firstly transferred on glass substrate by adhesive wafer bonding and deep reactive ion etching (DRIE). The thickness of the device layer is further reduced to 600 nm using inductively coupled plasma (ICP). To fabricate the metasurface pattern, a 300 nm-thickness hydrogen silsesquioxane (HSQ) layer is first spin-coated at 4000 rpm on the substrate and baked on a hot plate for 5 min at 90 °C. Then a 30 nm thickness aluminum layer (thermal evaporation) is deposited to serve as the charge dissipation layer. Next, the pattern is exposed using electron beam lithography (EBL). After exposure, the aluminum layer is removed by 5% phosphoric acid, and the resist is developed with

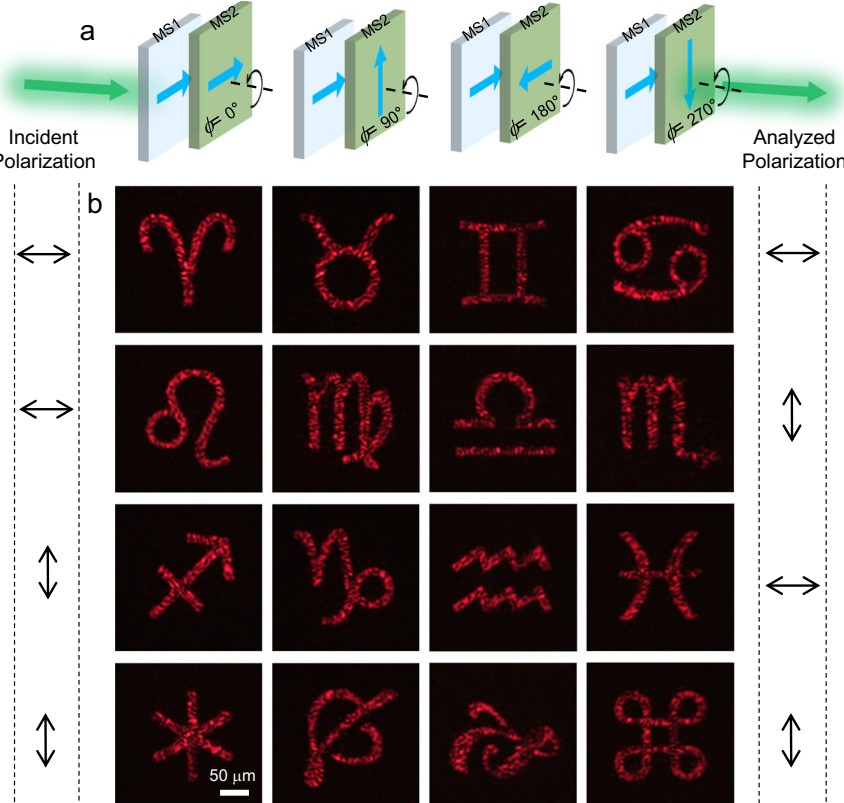

**Fig. 4 | Bilayer metasurface for polarization-rotation multiplexed multifunctional holography. a** Schematic of the two layer metasurfaces with different rotation angle $\phi$ = 0°, 90°, 180°, and 270°. The arrows on the metasurfaces show the relative rotation angles between the two layers. **b** Measured holographic images under the four different combinations of the incident and analyzed polarizations with rotation angles of $\phi$ = 0°(first row), 90°(second row), 180°(third row) and 270°(fourth row). The holographic images are designed at 1500 μm above the second layer in the glass substrate. The incident and analyzed polarizations are indicated at left side and right side, respectively. All panels share the same scale bar in the lower left one.

tetramethy- lammonium hydroxide. Finally, the sample is etched using ICP. An important note is that the two metasurfaces are cascaded front-to-front, and therefore the EBL pattern of the second metasurface should be flipped horizontally.

## Optical setup and measurement

A schematic of the optical setup for the experimental measurement is shown in supplementary fig. 13. A tunable laser source is used to generate the light beam with a wavelength of 808 nm. The laser source is collimated with uniform intensity in the center and obliquely incident on the bilayer metasurface. The different polarization is generated by a polarizer and a quarter waveplates (QWP) in front of two metasurface samples, which are separately mounted on 3D translational stages. The light scattered by the bilayer metasurface is collected by a 20×/0.50 objective and isolated with another pair of QWP and polarizer. To measure the optical images along oblique direction, we carry out a spatial filtering process in the Fourier plane. The Fourier plane of the objective lens (usually lies inside its barrel) is taken outside by adopting Lenses 1 and 2, which is then readily available for the filtering process. A continuously variable iris is placed at the Fourier plane to serve as the filter. The position and the diameter of the iris are determined according to designed parameters (See details in Supplementary section 9). The final image is focused by Lens 3 and formed on the CMOS camera.

## Data availability

The data underlying Fig. 2b of the Main Text are provided as Source data files. Any additional data that support this study are available from the corresponding author upon request. Source data are provided with this paper.

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

## Acknowledgements

This research was supported by the National Natural Science Foundation of China (92150107, 62075246), Guangdong Natural Science Funds (2022B1515020019) and Guangzhou Science and Technology Planning Project (202201010361). C.-W.Q. acknowledges financial support from the National Research Foundation, Prime Minister's Office, Singapore under Competitive Research Program Award NRF-CRP22-2019-0006. C.-W.Q. is also supported by a grant (A-0005947-16-00) from Advanced Research and Technology Innovation Centre (ARTIC).

## Author contributions

Y.B. conceived the idea, conducted the numerical simulations, fabricated the sample, performed the measurement and wrote the manuscript. X.Y., F.N., J.Y., and C.-W.Q. joined the discussions and gave useful suggestions. Y.B., C.-W.Q. and B.L. supervised the project.

## Competing interests

The authors declare no competing interests.
