## [Peer review file · Nature Communications]

REVIEWER COMMENTS

Reviewer #1 (Remarks to the Author):

The revision adequately addresses the comments from the previous review, and this work is now recommended for publication.

Reviewer #2 (Remarks to the Author):

The authors proposed a bilayer metasurface to realize the full parametric control of Jones matrix. The amplitude and phase of four units in Jones matrix, that is, eight degrees of freedom, can be modulated by this structure. Using optimized structures, the amplitude and phase of any nonorthogonal polarization components can be independently controlled, and 16-channel holograms with polarization-rotation multiplex multifunction can be realized. In general, the idea is interesting and the experiment results are convincing, and it can be published in Nature Communications if the authors can address my concerns:

- 1.The authors give examples to demonstrate the capability of complex amplitude modulation with nanoprinting in the near field and intensity distribution in the far field. The advantage of complex amplitude modulation lies in the precise control of the complex amplitude of the diffraction field, which is of great significance for both vector holography and 3D holography. What is the performance of this structure in the reconstruction of complex amplitude of the diffraction field? Because the nanoprinting results actually don't work out very well.
- 2.Following the previous question, as shown in Figure S10, the MSE of nanoprinting results is close to 1. What is the main reason?
- 3.How does it perform for the reconstruction of grayscale image?

Responses to Reviewers

Dear Editor,

We would like to thank you and the reviewers for taking the time to assess our paper. We highly appreciate the reviewers' constructive comments/suggestions on our manuscript for further improving the quality and clarity of our manuscript. Following the reviewers' comments/suggestions, we have improved our manuscript, and summarized our detailed responses and our revisions as follows.

Response to Reviewer#1

Reviewer #1 (Remarks to the Author):

The revision adequately addresses the comments from the previous review, and this work is now recommended for publication.

Reply: Thank the reviewer for the comments.

Response to Reviewer#2

Reviewer #2 (Remarks to the Author):

The authors proposed a bilayer metasurface to realize the full parametric control of Jones matrix. The amplitude and phase of four units in Jones matrix, that is, eight degrees of freedom, can be modulated by this structure. Using optimized structures, the amplitude and phase of any nonorthogonal polarization components can be independently controlled, and 16-channel holograms with polarization-rotation multiplex multifunction can be realized. In general, the idea is interesting and the experiment results are convincing, and it can be published in Nature Communications if the authors can address my concerns:

Reply: Thank the reviewer for the comments.

1.The authors give examples to demonstrate the capability of complex amplitude modulation with nanoprinting in the near field and intensity distribution in the far field.

The advantage of complex amplitude modulation lies in the precise control of the complex amplitude of the diffraction field, which is of great significance for both vector holography and 3D holography. What is the performance of this structure in the reconstruction of complex amplitude of the diffraction field? Because the nanoprinting results actually don't work out very well.

Reply: Thank the reviewer for the comments. We concentrate on the diffraction fields and design four holographic images with uniform phases. The Jones matrix now can be directly calculated by a reverse Rayleigh–Sommerfeld transformation of the input holographic images. In the figure below, we show a comparison between the complex holographic images and the previously calculated nanoprinting-holography. The much lower speckles and the calculated phase distributions (almost uniform as designed) show good reconstruction of complex amplitude in the diffraction field with our bilayer metasurface.

In Supplementary Materials, Page 23, we added “Similarly, if we set the phase of the holographic images to be uniform, the fidelity of image can be enhanced. In this case, the four Jones matrix now can be directly calculated by a reverse Rayleigh–Sommerfeld transformation of the input holographic images. Fig. S8 shows a comparison between the uniform-phase holographic images and the previously calculated nanoprinting-holography. The much lower speckles and the calculated phase distributions (almost uniform) show good reconstruction of complex amplitude in the diffraction field with

our bilayer metasurface.”

In Supplementary Materials, Page 25, we added Fig. S8, comparison between the complex holographic images and the previously calculated nanoprinting-holography.

2. Following the previous question, as shown in Figure S10, the MSE of nanoprinting results is close to 1. What is the main reason?

Reply: Thank the reviewer for the comments. We recheck our calculation and find that the problem arises from an inappropriate normalized ratio on the image intensities. The MSE is re-calculated and its dependence on the alignment shift is shown below. The MSE value of nanoprinting with zero shift is decreased to 0.51. Such value is still relatively high, because the nanoprintings present speckles and fringes due to the non-uniform phase distribution (destructive interference of adjacent pixels) and oblique detection. Thanks again for the concern from the reviewer.

In Supplementary Materials, **Page 29**, we added “Due to the non-uniform phase distribution (destructive interference of adjacent pixels) and oblique observation, the nanoprintings present speckles and fringes, making the MSE value relatively high.”

In Supplementary Materials, **Page 30**, we modified Fig. S12.

3. How does it perform for the reconstruction of grayscale image?

Reply: Thank the reviewer for the comments. When the nanoprinting is imaged, there are two problems: 1) The non-uniform phase distributions cause a highly speckly image due to the destructive interference of adjacent pixels. 2) The oblique observation cause

fringes, which is intendedly to eliminate the effect of the residual zero-order. We choose a three-level intensity image as a demonstration for the nanoprinting. In experiment and simulation, we can see a clear profile as the design ones but lost the details of the images. Therefore, it will be difficult to reproduce a grayscale 256-level image as design in nanoprinting.

There are some literatures that reproduce grayscale nanoprinting (Light: Science & Applications 7, 17129, 2018), but the phases must be uniform. To reconstruct a grayscale image with our bilayer metasurface, we choose a grayscale nanoprinting and set the phases of the four Jones matrix terms to be uniform as well. The simulated nanoprinting with FDTD is shown below, where the grayscale image is constructed very well. There are still some fringes in the image, which is mainly arisen from phases due to oblique detection.

In Supplementary Materials, Page 23, we added “As mentioned above, when a complex field is imaged, the phase distribution can cause a highly speckly nanoprinting image due to the destructive interference of adjacent pixels. Therefore, we only choose a three intensity level images as the nanoprinting for demonstration. If the phase distribution is set as uniform, grayscale nanoprinting image can be perfectly constructed (Fig. S7). The fringes arise due to the oblique detection. ”

In Supplementary Materials, Page 25, we added Fig. S7, the designed and simulated grayscale nanoprintings, where the phases of the Jones matrix are set as to be uniform for bilayer metasurface.